



# Scaling and balancing carbon dioxide fluxes in a heterogeneous tundra ecosystem of the Lena River Delta

Norman Rößger[1], Christian Wille[2], David Holl[1], Mathias Göckede[3], Lars Kutzbach[1]

[1] Institute of Soil Science, University of Hamburg, Allende-Platz 2, 20146 Hamburg
[2] German Research Centre for Geosciences, Telegrafenberg, 14473 Potsdam
[3] Max Planck Institute for Biogeochemistry, Hans-Knöll-Straße 10, 07745 Jena

Corresponding author: Norman Rößger (norman.roessger@uni-hamburg.de)

## Abstract

The current assessments of the carbon turnover in the Arctic tundra are subject to large uncertainties. This problem can (inter alia) be ascribed to both the general shortage of flux data from the vast and sparsely inhabited Arctic region, as well as the typically high spatiotemporal variability of carbon fluxes in tundra ecosystems. Addressing these challenges, carbon dioxide fluxes on an active flood plain situated in the Siberian Lena River Delta were studied
during two growing seasons with the eddy covariance method. The footprint exhibited a heterogeneous surface, and the mixed flux signal associated therewith could extensively be decomposed: respiratory loss and photosynthetic gain were not only modelled for the overall footprint, but also for each of two vegetation classes. This downscaling of the observed fluxes unveiled a differing seasonality in the net uptakes of bushes ($-0.89$ $\mu$mol m$^{-2}$ s$^{-1}$) and sedges
($-0.38$ $\mu$mol m$^{-2}$ s$^{-1}$) in 2014. That discrepancy, which was concealed in the net signal, resulted from a comparatively warm spring in conjunction with an early snow melt and a varying canopy structure. Thus, the representativeness of footprints may adversely be affected in response to prolonged unusual weather conditions. In 2015, when air temperatures on average corresponded to climatological means, both vegetation class-specific flux rates were of similar mag-
nitude ($-0.69$ $\mu$mol m$^{-2}$ s$^{-1}$). A comprehensive set of measures (e.g. phenocam) approved the reliability of the partitioned fluxes, and hence confirmed the utility of the flux decomposition for enhanced flux data analysis. This scrutiny encompassed insights into both the phenological dynamic of individual vegetation classes, plus their respective functional flux to flux driver relationships with the aid of ecophysiologically interpretable parameters. For the purpose of
comparison with other sites, the decomposed fluxes were employed in a vegetation class area-weighted upscaling that was based on a classified high-resolution orthomosaic of the flood plain. In this way, robust budgets that take the heterogeneous surface characteristics into account were estimated. In relation to the average sink strength of various Arctic flux sites, the flood plain constitutes a distinctly stronger carbon dioxide sink. Roughly 42 % of this net up-
take, however, was on average offset by methane emissions lowering the sink strength for greenhouse gases. With growing concern about rising greenhouse gas emissions in high-latitude regions, providing robust carbon budgets from tundra ecosystems is critical in view of the thawing permafrost, whose released carbon can impact the global climate for centuries.



## 1. Introduction

Permafrost underlies between 12.8 % and 17.8 % of the exposed land area in the northern hemisphere (Zhang et al., 2000). Large parts of this area coincide with the Arctic tundra, which is situated north of the boreal treeline and covers roughly 8 % of the global land surface (McGuire et al., 2012). As a consequence of the historical carbon sink function, the underlying permafrost forms a carbon stock of global relevance: approximately 1300 Gt soil organic carbon are stored in the circumpolar permafrost region (Hugelius et al., 2014). However, large fractions of this carbon pool may be remobilised in response to a warming climate making the tundra a key ecosystem for climate change (Schuur et al., 2008).

The Arctic north of 60° N has warmed at a rate of 1.36 °C per century since 1875, i.e. roughly twice as fast as the global average (Masson-Delmotte et al., 2013). And the rapid warming trend is projected to continue (Collins et al., 2013). However, due to ambiguous results and large confidence intervals, it currently remains unclear, whether the permafrost areas maintain their long-term sink function or convert into a carbon source in the future (Heimann and Reichstein, 2008; Schuur et al., 2015). These uncertainties do not only arise from the limited knowledge on the physical thawing rates, the fraction of released carbon after thawing and the time scales of release, but also from the general shortage of flux data in Arctic ecosystems (Ciais et al., 2013). The scarce data availability particularly applies to the extensive Siberian tundra, which covers around 3 million $km^2$, i.e. more than half of northern high-latitude tundra ecosystems (Chapin III et al., 2005; Sachs et al., 2010). The low density of flux observation sites is due to both harsh environmental conditions as well as challenging logistics in these remote and sparsely inhabited areas often without line power. Consequently, current estimates of the tundra's sink strength for carbon dioxide are associated with large ties: -103 ± 193 Tg C $yr^{-1}$ (McGuire et al., 2012). The same issue applies to estimates that suggest a shift to a source for carbon dioxide: 462 ± 378 Tg C $yr^{-1}$ (Belshe et al., 2013). The reduction of these discrepant uncertainties (concerning sign and magnitude) can be achieved via providing more carbon budgets from the Siberian tundra as well as a more reliable characterisation of the variation in habitats (e.g. bogs, fens,) plus their associated surface heterogeneity (e.g. tussocks, hummocks).

Tundra ecosystems are frequently characterised by a pronounced vegetation patchiness with sharply defined boundaries between differing vegetation classes (Shaver et al., 2007). The consequently high spatial variability in carbon fluxes aggravates the estimation of robust carbon budgets that are accurate and precise. Therefore, a better understanding of the effects of surface heterogeneity on these budgets, e.g. through a better characterisation of both flux variability and associated environmental controls such as vegetation composition and structure, is necessary (Kade et al., 2012; Kwon et al., 2006).

Chamber measurements operate on the microscale ($10^{-2}$-$10^2$ $m^2$), and hence form a suitable approach to differentiate the carbon dioxide exchange of multiple microforms with the atmosphere (McGuire et al., 2012). However, chamber measurements are associated with several drawbacks such as (i) a disturbance of the studied system, (ii) a mostly discontinuous sampling, (iii) a limited spatial representativeness, (iv) a decoupling of the sampled surface from the atmosphere that causes a modification of the environmental conditions (e.g. temperature,





humidity, radiation, wind speed, air pressure) in the headspace, and (v) an alteration of the concentration gradient across the soil-atmosphere interface that inherently impacts on the diffusive soil gas transport during chamber deployment (Fox et al., 2008; Kade et al., 2012; Kutzbach et al., 2007a; Livingston et al., 2006; Riederer et al., 2014; Sachs et al., 2008;

Wagner and Reicosky, 1992).

Alternatively, non-intrusive and continuous eddy covariance measurements, which operate on the mesoscale ($10^4$-$10^6$ m$^2$), yield turbulent fluxes that integrate across multiple microforms (Aubinet et al., 2012). The size and location of the sampled surface constantly shifts according to wind direction, wind speed, atmospheric stability, crosswind velocity and surface roughness

(Detto et al., 2006). In the presence of a heterogeneous landscape, the temporal variability in the observed fluxes is not only a result of the varying uptake/release rates of the individual microforms, but also an outcome of the varying fractions of microforms in the sampled area. In addition, the footprint budgets may lack representativeness since the fractional composition of microforms within the footprint is likely to deviate from the microform distribution in the area

of interest. In such an environment, budgets strongly depend on tower location, sensor height and wind field conditions, and are thus likely to exhibit a sensor location bias (Schmid and Lloyd, 1999). Moreover, heterogeneous flux signals also aggravate an appropriate determination of model parameters, e.g. the light response curve of a vegetation type, if the corresponding microforms exhibit strongly deviating characteristics (Lasslop et al., 2010). Despite these

challenges in signal interpretation, a heterogeneous surface also provides the opportunity to conduct a concurrent sampling of multiple microforms and the study of their carbon dioxide fluxes utilising only one eddy covariance instrumentation (Forbrich et al., 2011; Morin et al., 2017). Exploiting this potentially valuable information source requires the partitioning of the integrated flux into its microform-specific fluxes. Such a successful flux decomposition routine

yields microform-specific budgets that, in conjunction with a precise determination of the microforms' spatial coverages in the area of interest, enable the estimation of robust carbon dioxide budgets for a heterogeneous surface.

Addressing the problems of balancing carbon fluxes in a Siberian tundra ecosystem with both a heterogeneous surface and an unknown greenhouse gas sink/source strength, the objectives

of this study are as follows: (i) analysing the spatiotemporal variability of carbon dioxide fluxes utilising both the eddy covariance technique and footprint modelling, (ii) elucidating the heterogeneity of the footprint and its impact on the flux dynamics, (iii) estimating robust carbon dioxide budgets that account for the heterogeneity of the landscape, and (iv) combining these budgets with previously estimated methane budgets in order to determine the

sink/source strength for greenhouse gases.

## 2. Material and Methodology

### 2.1. Site description

The Lena River Delta, one of the largest deltas in the world, is located within the zone of con-

tinuous permafrost in northern Siberia (Fig. 1). One of its numerous islands is Samoylov Island (72° 22' N, 126° 28' E), which covers an area of 4.8 km$^2$ and features two geomorphological units: the late-Holocene river terrace in the eastern part and the active flood plain in the





western part. The carbon dioxide exchange on the river terrace, which is characterised by ice-wedge polygonal tundra with sedges and mosses, has been repeatedly studied (Eckhardt, 2018; Kutzbach et al., 2007b; Runkle et al., 2013). In contrast to the river terrace, the flood plain has to date received scarce attention in terms of greenhouse gas fluxes although active flood plain levels represent roughly 40 % of the soil-covered area of the Lena River Delta (Zubrzycki et al., 2013). Aside from the period of the annual spring flood, whose associated inundation is very variable in magnitude and duration, the flood plain on Samoylov Island stretches over an area between 1 km$^2$ (spring) and 2 km$^2$ (autumn). More importantly, the surface of the flood plain exhibits, in opposition to the river terrace, a distinct heterogeneity on the mesoscale.

The central delta region is situated in a continental Arctic climate, which is characterised by very low temperatures and a low annual precipitation. In the distant town of Tiksi, located around 120 km southeast of Samoylov Island, a mean annual air temperature of -12.8 °C was measured during 1936-2016 and a mean annual precipitation of 329 mm was gauged during 1956-2016 (AARI, 2017). Additional information on this study site can be found in Rößger et al. (2018).

### 2.2. Experimental setup and data recording

An eddy covariance system was installed in the southern part of the flood plain, and the measurements covered two periods: 18$^{th}$ June to 2$^{nd}$ October 2014 (107 days) and 9$^{th}$ June to 24$^{th}$ September 2015 (108 days).

The flux tower was equipped with a sonic anemometer (CSAT3, Campbell Scientific, UK) and a gas analyser for water vapour and carbon dioxide (LI-7500A, LI-COR Biosciences, USA). Both instruments were mounted at a height of 2.83 m, and sampled with a frequency of 20 Hz. In addition, another eddy covariance system with the same instrumentation has already been deployed at a central position on the adjacent river terrace (Holl et al., 2018).

Supplementary measurements on the flood plain involved acquiring data of both air temperature (HMP45, Campbell Scientific, UK) and photosynthetic photon flux density (SKP215, Skye Instruments, UK), These environmental variables were recorded on a logger (CR1000, Campbell Scientific, UK) in a quarter-hourly interval. Furthermore, a time lapse camera (TLC200, Brinno, Taiwan) was mounted on the flux tower pointing towards northeast for monitoring the phenology during spring 2014 with the same interval of an quarter of an hour.

### 2.3. Flux processing

The flux computation was carried out with the software EddyPro version 6.0.0 (LI-COR Biosciences, 2016) for 30-min flux intervals, and followed the standard procedure. Detailed information on the executed (i) raw data processing (spike removal, tilt correction, block averaging, time lag compensation), and the implemented (ii) flux correction scheme (WPL correction, spectral correction in low and high frequency range, flux error estimation), and the conducted (iii) quality assessment routine (stationarity test, integral turbulence characteristics test, skewness and kurtosis examination, energy flux quality verification, signal strength control, percentile removal) is provided in Rößger et al. (2018).



For the footprint modelling, an analytical model for non-neutral stratification was employed (Kormann and Meixner, 2001). This model is based on a stationary gradient diffusion formulation with height-independent crosswind dispersion (Leclerc and Foken, 2014). Applying the solution of the resulting two-dimensional advection-diffusion equation for solving the power law profiles of both eddy diffusivity and mean wind velocity, it yielded a source weight function for each flux interval.

## 2.4. Surface structure

For studying the impact of the heterogeneous surface on the flux variability, the entire flood plain was mapped in August 2014 employing helicopter-based visible aerial imagery. The resulting geo-referenced orthomosaic exhibited a resolution of 8.5 cm, and hence provided a very high spatial information density, sufficient to resolve the pronounced heterogeneity of the surface. Based on maximum likelihood classification tools, the vegetation was classified employing a supervised classification routine on the orthomosaic (Fig. 2). In this process, four different land cover classes were utilised, two of which represent the vegetation.

Vegetation class 1 ("bushes") refers to sites, which were densely vegetated by large dwarf shrubs of the willow family such as S*alix pulchra*, *Salix lanata, Salix hastata, Salix glauca*, growing to a maximum height of around 1 m. This shrubby vegetation was located on a sandy ridge aligned in the north-south-axis. The elevated area enabled, in conjunction with a spatially averaged maximum thaw depth of 0.93 m, a good drainage. Since the ground water table remained at depths around 50 cm, the surface was mostly dry, forming favourable growing conditions for willow shrubs and a sparse cover of thin moss.

Vegetation class 2 ("sedges") represents areas, which were dominated by sedges including *Carex aquatilis*, *Carex chordorrhiza*, *Carex concolor* as well as species of *Eriophorum* and *Equisetum*. Also, small willow shrubs growing to a height of about 0.3 m were occasionally to be found. This predominantly graminoid vegetation was located in depressions around the dry ridge with a mean active layer depth of 0.69 m. Accordingly, the soil moisture conditions alternated between moist surfaces and wet patches with water levels up to 40 cm. The ample moisture attracted many mosses forming a dense cover of tick moss.

The two other classes, which do not occur in the 90 % contribution footprint around the flux tower, denote a large area of bare sand along the waterfront and some small water bodies mainly situated in the northern part of the flood plain (Fig. 2). The former class was not considered in the budget estimation since its carbon dioxide flux rates were (in comparison to the two vegetation classes) assumed negligible. The latter class was appended to vegetation class 2 as the few small water bodies surrounded by sedges were presumed to have similar flux rates. Further information on the classification routine are given in Rößger et al. (2018).

## 2.5. Flux modelling

The model structure is based on the computation of the two components of the carbon dioxide flux.

$$F_{CO_2} = NEE = TER - GPP \tag{1}$$



$F_{CO2}$ is the net carbon dioxide flux observed at the flux tower, and equals NEE, the net ecosystem exchange. Its two components TER and GPP describe, respectively, the total ecosystem respiration and the gross primary productivity, both of which can be modelled simultaneously (Runkle et al., 2013).

$$NEE = R_{base} \cdot Q_{10}^{\left(\frac{T_{air}-T_{ref}}{\gamma}\right)} - \frac{P_{max} \cdot \alpha \cdot PPFD}{P_{max}+\alpha \cdot PPFD} \tag{2}$$

The parameter $R_{base}$ denotes the basal respiration at the reference temperature ($T_{ref}$), which was set to 15 °C, and a scaling factor ($\gamma$) was held constant at 10 °C (Mahecha et al., 2010). $Q_{10}$ indicates the temperature sensitivity; i.e. this parameter is a value by which respiration multiplies/divides, when the temperature rises/drops by 10 °C. The parameter $P_{max}$ refers to

the maximum photosynthetic potential and quantifies the theoretical maximum of photosynthesis at infinite irradiance. $\alpha$ represents the light sensitivity and states, as the initial quantum efficiency, the slope of the light response curve at irradiance being zero. All of the four (physiologically interpretable) parameters are best-fit parameters, which were estimated via nonlinear ordinary least-squares regression utilising both air temperature ($T_{air}$) and photosynthetic

photon flux density (PPFD) as explanatory variables. In order to take the heterogeneous surface structure into account, footprint information were included forming the final model employed for carbon dioxide flux modelling.

$$NEE = \sum_{i=1}^{i=2} \Omega_i \cdot \left( R_{base,i} \cdot Q_{10,i}^{\left(\frac{T_{air}-T_{ref}}{\gamma}\right)} - \frac{P_{max,i} \cdot \alpha_i \cdot PPFD}{P_{max,i}+\alpha_i \cdot PPFD} \right) \tag{3}$$

Another explanatory variable is the relative contribution of each vegetation class to the flux ($\Omega$) that weights the two computed vegetation-specific fluxes. This variable was obtained through (i) computing the source weight function for a flux interval, (ii) spatially discretising this continuous function with a resolution of 1 m, (iii) adjusting the vegetation map to a resolution of 1 m, (iv) assigning each value of the source weight function to its spatially corre-

sponding vegetation class, and (v) summing the values in each vegetation class.

The fitting procedure, in which only half-hourly quality-controlled flux data was employed, required the estimation of a large number of fitting parameters: $R_{base}$, $Q_{10}$, $P_{max}$, $\alpha$ for each vegetation class (Fig. 3). In order to avoid overparameterisation and equifinality problems, the model structure was gradually simplified along four different steps. These alterations in the

parametrisation enabled the desired estimation of (i) reasonable seasonal courses of the fitting parameters, i.e. courses that displayed a predominantly smooth evolution with elevated values during the growing season and low values in the shoulder seasons, and (ii) meaningful and significant values for the fitting parameters, i.e. values that were not negative as well as within an acceptable range and their 95 % confidence interval did not overlap zero. Achieving both

objectives provided the possibility to interpret the fitting parameters ecophysiologically.

In each of the four parameterisation steps, the respectively parametrised model was recalibrated for every day applying a moving window with fixed/flexible window sizes and a step size of one day. In the initial step 1, which served the computation of representative $Q_{10}$ values, all of the eight fitting parameters were estimated in the model (4-4-p). Through its output, which

encompassed eight best-fit time series, a representative $Q_{10}$ value was obtained for each vege-




tation class by determining the median out of the best-fit $Q_{10}$ values that fulfilled two requirements: statistical significance and an associated coefficient of determination (between observed NEE and modelled NEE) of 0.75 or greater. These two $Q_{10}$ values were held constant in the further fitting procedure. In the subsequent step 2, the simplified model (3-3-p) was run

with six fitting parameters to be estimated, and a Gaussian bell curve was fitted to the time series of significant best-fit α values for each vegetation class. By adding/subtracting 30 % of the function values to/from these two replacement function itself, a pair of encompassing threshold functions was respectively appended. These intervals around the replacement functions formed a range, inside which best-fit α values were accepted. In the following step 3, the

model (3-3-p) was run with the same parametrisation of the previous step. The model output was checked for α values inside the acceptable interval as well as significant values for $R_{base}$, $P_{max}$ and α. If these criteria were satisfied, the accordingly modelled NEE was approved and the fitting procedure proceeded to the next day. Alternatively, several models (3-2-p/2-3-p/2-2-p) were run employing α value(s) from the replacement function(s) for one or both vegeta-

tion classes, depending on which vegetation class insignificant and/or implausible best-fit parameters were in. The output was tested again, and in case of significant best-fit parameters, the modelled NEE was accepted and the fitting procedure continued with the next day. Lastly, a replacement function for $P_{max}$ was created by fitting a Gaussian bell curve to the time series of significant best-fit $P_{max}$ values in both vegetation classes. In the final step 4, two

greatly simplified models (2-1-p/1-2-p) were run with only three fitting parameters to be estimated as well as a $P_{max}$ value from the replacement function. Here, if not before, all fitting parameters have taken on meaningful and significant values, which ensured the computation of reliable NEE values. In addition to this brief explanation, a detailed description of the entire fitting process is attached in the appendix.

Since the model was designed to simultaneously compute the component fluxes in both vegetation classes, it provided the capability for the decomposition of the observed fluxes into their separate flux contributions by the two vegetation classes. The reliability of this downscaling, however, was dependent on the restrictive acceptance of meaningful and significant values for the fitting parameters. The temporal integration of these partitioned fluxes, and the subse-

quent projection of the resulting budgets on their corresponding areas on the flood plain formed the upscaling. The summation of both vegetation class budgets finally yielded a robust budget of the entire flood plain, which was designated as the area of interest. This budget, as opposed to the directly estimated footprint budget, did not exhibit a sensor location bias, and hence allowed an unbiased appraisal of both the interannual variability and the sink/source

strength. For the sake of comparability of the budgets between the years and with already estimated methane budgets, carbon dioxide budgets were calculated for the comparison period 18[th] June to 24[th] September, where data was available in both years.



## 3. Results

### 3.1. Meteorological conditions

The mean air temperatures during the measurement periods in 2014 and 2015 amounted to 7.7 °C and 7.1 °C, respectively. Furthermore, the respective precipitation sums totalled
92.3 mm and 130.4 mm. The assessment of these values was based on their comparison with long-term averages that were obtained for Samoylov Island with records from 1998 to 2018 (Fig. 4). The measurement period in 2014 was on average distinctively warmer and slightly drier, while the measurement period in 2015 featured the same mean temperature as the baseline, but considerably more rain. The largest differences in air temperature between both
measurement periods occurred in spring. Accordingly, the snow melt in 2014 took place in a prolonged manner during mid-May already, whereas in 2015, the snow melt was completed within a couple of days in early June, as usual.

### 3.2. Dynamics of observed fluxes

The carbon dioxide fluxes exhibited both a diurnal and a seasonal course with the following mean fluxes that were obtained by averaging half-hourly flux data for the sub-seasons in both years 2014 and 2015 (Fig. 5). Between the snow melt and the vegetative phase, the mean carbon dioxide fluxes remained slightly positive, indicating a dominating respiration while the vegetation largely remained dormant (0.26 $\mu$mol m$^{-2}$ s$^{-1}$). With the onset of the growing season
in late June, stalks and foliage began to develop, and the uptake of carbon dioxide during daytime outweighed the release of carbon dioxide during nighttime (-1.06 $\mu$mol m$^{-2}$ s$^{-1}$). The intensity of this oscillation increased towards the onset of the reproduction phase in mid-July, where flowers and seeds developed. During this phase, the most negative fluxes occurred featuring a relatively constant magnitude (-1.77 $\mu$mol m$^{-2}$ s$^{-1}$). With the onset of the ripening
phase in early August, bushes and sedges verged on full maturity, and the flux amplitude of the diurnal cycle began to be progressively attenuated (-0.78 $\mu$mol m$^{-2}$ s$^{-1}$). During the nights of this period, the most positive fluxes occurred. Towards late August, the respiration exceeded photosynthesis again indicating the onset of the senescence phase, which was associated with both colouration and shedding of leaves (0.39 $\mu$mol m$^{-2}$ s$^{-1}$). After the end of the growing
season in early September, when abscission was completed, the dominance of respiration continued to grow, leading to more positive mean carbon dioxide fluxes (0.55 $\mu$mol m$^{-2}$ s$^{-1}$).

### 3.3. Model calibration and performance

While the $Q_{10}$ values were optimised at constant values of 1.42 for vegetation class 1 and 1.48
for vegetation class 2, the other fitting parameters $R_{base}$, $P_{max}$ and $\alpha$ displayed a seasonal course for each vegetation class in 2014 and 2015 (Fig. 6 and Fig. 7). The temporal evolution of $\alpha$ values could be well approximated with replacement functions, whose application reduced the noise not only in the seasonal courses of $\alpha$, but also in the seasonal courses of both $R_{base}$ and $P_{max}$. In contrast to the replacement functions of $\alpha$, which were created for both vegeta-
tion classes in both years, a replacement function for $P_{max}$ was created only for vegetation class 1 in 2015 and for vegetation class 2 in 2014.





The slightly simplified 3-3-p model, which was run at the start of step three, yielded meaning-ful and significant values for the fitting parameters in 49 % of the modelled days including 2014 and 2015 (Fig. 3). In the further course of step three, these goals were achieved by the gradually simplified 3-2-p/2-3-p/2-2-p models in 47 % of the modelled days. During the re-
maining 4 %, the greatly simplified 2-1-p/1-2-p models of step four were deployed. While the 3-3-p model was mainly employed during the summer season, the 3-2-p/2-3-p/2-2-p models were applied throughout the measurement periods with a focus on the shoulder seasons. The 2-1-p/1-2-p models were solely deployed during the shoulder seasons and more often during spring than during autumn. Hence, larger fluxes during the growing season could be more easi-
ly modelled in comparison to the remaining time, when lower fluxes associated with a less fa-vourable signal-to-noise ratio prevailed.

$R_{base}$ was the fitting parameter that could be estimated most confidently as this parameter ac-counted for only 19 % of the insignificant values obtained during the fitting procedure in 2014 and 2015. While $P_{max}$ caused 31 % of the insignificances, $\alpha$ appeared to be the least certain fit-
ting parameter representing the remaining 50 %. Furthermore, the best-fit $P_{max}$ values of both vegetation classes featured most of the significant differences between each other, i.e. the con-fidence intervals of both vegetation classes rarely overlapped, whereas best-fit $\alpha$ values exhibit-ed the fewest significant differences.

On account of both the coinciding variabilities of explanatory variables and explained variable
as well as the recalibration for each day, the model was able to reproduce the observed fluxes very well (Fig. 5). This performance is expressed by coefficients of determination ($R^2$) of 0.88 for 2014 and 0.95 for 2015. Furthermore, the mean absolute errors (MAE) amounted to 0.49 $\mu mol\ m^{-2}\ s^{-1}$ and 0.35 $\mu mol\ m^{-2}\ s^{-1}$ for 2014 and 2015, respectively, while the root mean square errors (RMSE) amounted to 0.75 $\mu mol\ m^{-2}\ s^{-1}$ and 0.52 $\mu mol\ m^{-2}\ s^{-1}$. The model per-
formed better during the summer season and less good during the shoulder seasons, where au-tumn displayed a slightly better performance than spring.

### 3.4. Downscaling and upscaling of fluxes

The assignment of individual parameter sets in the model allowed the decomposition of the
observed net fluxes. This downscaling yielded fluxes of NEE plus their component fluxes TER and GPP accounting for both vegetation classes in both years (Fig. 8). For the comparison pe-riod in 2014, the mean NEE amounted to -0.89 $\mu mol\ m^{-2}\ s^{-1}$ and -0.38 $\mu mol\ m^{-2}\ s^{-1}$ for vegeta-tion class 1 and vegetation class 2, respectively, and for the comparison period in 2015, -0.71 $\mu mol\ m^{-2}\ s^{-1}$ and -0.69 $\mu mol\ m^{-2}\ s^{-1}$ (Table 1). In contrast to the similar mean net
uptakes in 2015, the mean net uptakes in 2014 distinctly differed from each other. This dis-crepancy originated from the first half of the growing season (mid-June to early August), when the net uptake of vegetation class 1 was considerably larger relative to vegetation class 2. Dur-ing the second half of the growing season (early August to late September), both net uptakes were rather similar again. Furthermore, the differences in the net uptakes between both years
were governed by changes in GPP rather than in TER. In vegetation class 1, NEE in 2014 was only slightly greater in comparison to 2015, which can be attributed to a greater TER and a distinctly greater GPP. And in vegetation class 2, NEE in 2014 was smaller compared to 2015, which can be ascribed to a smaller TER and a clearly smaller GPP.



The aggregation of the decomposed fluxes over the comparison period yielded individual budgets, whose multiplication with the corresponding fractional coverages on the flood plain formed the upscaling (Table 1). The subsequent summation of both vegetation class-specific net uptakes returned the net uptake of the entire flood plain for the comparison period: -4.42 ± 0.49 Mmol in 2014 and -6.17 ± 0.66 Mmol in 2015. The stated uncertainties were obtained by means of standard error propagation techniques including both cumulative flux error and classification error, where the former was one magnitude smaller than the latter. Dividing these budgets by the total area of the flood plain yielded mean flood plain budgets of -4.22 ± 0.47 mol m$^{-2}$ and -5.89 ± 0.63 mol m$^{-2}$ (Table 2). These budgets consider the surface heterogeneity, i.e. they are corrected for the sensor location bias, plus they contain an areal reference, and thus enable an appropriate comparison with other sites.

### 3.5. Greenhouse gas balances

The evaluation of the flood plain's sink/source strength for greenhouse gases required the corresponding methane emission budgets and their conversion to carbon dioxide equivalents (Rößger et al., 2018). Despite the methane's minor percentage of roughly 3 % in the entire greenhouse gas exchange (specified in molar units), its carbon dioxide equivalents diminished the greenhouse gas sink strength (given by the carbon dioxide net uptake) by half in 2014 and by one third in 2015. Accordingly, the greenhouse gas balances specify that the flood plain formed a moderate sink of -2.21 ± 0.61 mol $CO_2$ eq m$^{-2}$ and a stronger sink of -3.81 ± 0.74 mol $CO_2$ eq m$^{-2}$ during the warm season in 2014 and 2015, respectively (Table 2). The lower sink strength in 2014 was a result of a reduced carbon dioxide net uptake rather than an augmented methane efflux. And this reduced carbon dioxide net uptake in turn was caused by a lowered net uptake in vegetation class 2 that effectively counteracted the elevated early season net uptake in vegetation class 1. This class constituted a stronger greenhouse gas sink than vegetation class 2 in both years, which is mainly due to the fact that methane emissions were only present in vegetation class 2. Since these emissions hardly changed between the years as well as the negligible methane release in vegetation class 1, the interannual variability in the greenhouse gas sink strength was governed by the carbon dioxide net uptake.

These balances are the first greenhouse gas budgets of a flood plain in the Lena River Delta. Based on these budgets, the sink strength of the adjacent river terrace, where another eddy covariance system has been in operation for many years, could finally be put in context within the domain of the Lena River Delta (Table 2). In 2014 and 2015, the flood plain sequestered per square metre roughly 20 % and 60 % more carbon dioxide, respectively, but it also emitted approximately 70 % more methane. Hence, the flood plain constituted a sink for greenhouse gases that resembled (2014) or was 1.5 times (2015) the sink strength of the polygonal tundra on the river terrace.

### 4. Discussion

### 4.1. Assessment of the flux decomposition model

The partitioning of carbon dioxide fluxes was conducted during the Arctic summer, when fully dark conditions during the nights are absent. Consequently, a partitioning approach that is





based on fitting parameters to nighttime respiration followed by extrapolating these fits to daytime and subsequently subtracting the estimated daytime respiration flux from the observed net flux to obtain the photosynthesis flux is confronted with elevated uncertainties (Reichstein et al., 2005). The partitioning approach of the present study avoids this problem

since the parameter fitting employs the entire dataset. However, the model may have a shortcoming in the small number of environmental driving parameters, which may oversimplify the complex biogeochemical processes involved in the carbon dioxide exchange between soils, plants and the atmosphere. While the entire temperature sensitivity of NEE is manifested through changes in TER, the effect of temperature on the biochemical reactions in GPP is ne-

glected (Haraguchi and Yamada, 2011). At the same time, no correlation between air temperature and model residuals (between observed and modelled NEE) could be detected, which indicates that the temperature-induced variability was sufficiently considered. The confounding effect of a high vapour pressure deficit (VPD), which tends to take place in the afternoon leading to a limited photosynthetic activity, was not taken into account (Lasslop et al., 2010).

However, only very few days with low humidity (VPD>10 hPa) occurred, and the typically asymmetric diurnal cycle of NEE could not be found on these days. A missing linkage of the model with potential flux limitations through a low soil moisture is deemed appropriate given the constantly high moisture availability in the permafrost-affected soils at the study site (Gao et al., 2017; Minkkinen et al., 2018). The diverse effect of direct and indirect radiation on pho-

tosynthetic efficiency was also not taken into consideration (Williams et al., 2014). This effect plays a tangential role for the low sedges, but adds uncertainty to the light-response curves calculated for the larger shrubs. Further uncertainty may also be appended by a potential inaccuracy in both surface classification and footprint model. While the former is deemed appropriate due to extensive ground truthing, the latter is difficult to assess. However, the employed

footprint model is a widely applied tool within the flux community, and it constitutes a suitable model for this study site in a flat tundra landscape with low roughness lengths (Foken, T., personal communication, 2015). More importantly, the flux decomposition method, as carried out in the present study, may approach methodical limits, if the surface classes in the footprint are too uniformly distributed and/or their individual flux rates are too similar. Whether

the assignment of flux rates from a mixed signal to individual surface classes is still possible under these circumstances may be an objective of further studies at other sites.

## 4.2. Validation of the decomposed fluxes

The flux decomposition yielded insights into the flux dynamics of both investigated vegetation

classes. The validity of these dynamics and hence the reliability of the employed model is examined utilising four approaches.

Firstly, it has been demonstrated that the photosynthetic cycle of a canopy during a growing season is linked to its seasonal changes in greenness (Peichl et al., 2014; Sonnentag et al., 2012). The evolution of canopy greenness can be examined by determining the green chromatic

coordinates ($g_{cc}$) of a target area in images obtained by digital repeat photography (Richardson, 2012). Employing the images from the time lapse camera on the flux tower, this method yielded $g_{cc}$ values for vegetation class 1 with a central tendency that is significantly greater than the one of the $g_{cc}$ values for vegetation class 2 ($P<0.05$). These differences in



greenness substantiate the most prominent result of the flux decomposition: the greater photosynthesis of vegetation class 1 at the onset of the growing season 2014 (Fig. 9).

Secondly, during periods with a certain wind direction and atmospheric stability, the fetches of some observed fluxes were strongly dominated by only one vegetation class as opposed to the commonly mixed signals. Thus, observed fluxes that are accompanied with a large contribution of one vegetation class ($\Omega>0.7$) were compared to fluxes that were modelled for the same vegetation class. The choice of an $\Omega$ of 70 % rested in the desire to identify a trade-off between both applying many fluxes for a broad statistical basis (low $\Omega$) and utilising many fluxes without a mixed fetch for an accurate evaluation (large $\Omega$). Both observed and modelled fluxes match very well as indicated by a mean $R^2$ of 0.88 and a mean RMSE of 0.82 $\mu$mol m$^{-2}$ s$^{-1}$. Putting these values in context, in a study, where NEE of shrubs and sedges in tundra landscapes was modelled with non-linear regression, a mean $R^2$ and mean RMSE of 0.69 and 2.15 $\mu$mol m$^{-2}$ s$^{-1}$ was respectively obtained (Shaver et al., 2007). The decomposed fluxes of the present study are, when MAE is applied as an intuitive error metric, associated with a mean error of roughly 0.56 $\mu$mol m$^{-2}$ s$^{-1}$. The frequent similarity of the vegetation-class specific flux rates, however, reduces the effectivity of this validation test. Therefore, the observed fluxes governed by one vegetation class were also compared to fluxes modelled for the other class. This counter-check caused a rise in mean RMSE and MAE by 89 % and 99 %, respectively, thus lending further credibility to the modelled flux rates. It can be assumed that this rise would be far greater, if the flux rates of both vegetation classes were less similar.

Thirdly, closed chamber measurements have been carried out with an opaque chamber during mid-June 2014 in vegetation class 2 east of the flux tower (Runkle, B. and Sabrekov, A., personal communication, 2016). Similar to the respiration modelled for this class, a mean carbon dioxide flux with a standard deviation of 2.1 ± 0.9 $\mu$mol m$^{-2}$ s$^{-1}$ was observed. This mean, however, is based on 5 chamber measurements, and thus conclusive to only a limited extent since taking the spatial variability into account is crucial, when fluxes are scaled between eddy covariance and chamber measurements (Oechel et al., 1998). A great deal of the studies, which are concerned with upscaling chamber-derived fluxes in heterogeneous environments, are challenged (besides the typical downsides during the measurement) by the following problems: (i) a subjectivity in the selection of chamber locations, (ii) a low spatial representativeness due to both the small sampled size and only a few replicate sites as a result of a high labour intensity, (iii) a lacking acquisition of a pronounced temporal flux variability on account of a usual confinement to discrete sampling, and (iv) the accordingly numerous gaps in the time series that are dominated by modelled instead of observed values after the gap filling (Fox et al., 2008; Heikkinen et al., 2002; Kade et al., 2012; Laine et al., 2006; Marushchak et al., 2013).

Fourthly, the discussion and comparison of the obtained fitting parameters with values estimated at other sites gives further confidence in the validity of the decomposed fluxes (Fig. 6 and Fig. 7):

➢ The estimated $R_{base}$ values follow a temperature-driven seasonal cycle, in which $R_{base,2}$ is mostly lower than $R_{base,1}$. A smaller autotrophic respiration can be attributed to the lesser biomass of the sedges, and a smaller heterotrophic respiration can be ascribed to both increased soil moisture and decreased soil temperature, which in turn hamper microbial ac-





tivity in the depressions (Hobbie et al., 2000; Walz et al., 2017). For comparison with values found at other sites, a mean peak season TER was computed for vegetation class 1 (2.8 $\mu$mol m$^{-2}$ s$^{-1}$) and vegetation class 2 (2.3 $\mu$mol m$^{-2}$ s$^{-1}$). While the latter respiratory rate corresponds to the mean mid-growing season TER of 2.2 $\mu$mol m$^{-2}$ s$^{-1}$, which was estimated for northern peatlands, the former rate is greater (Frolking et al., 1998; Laurila et al., 2001). The comparatively large respiration in vegetation class 1 is likely due to both the large willow shrubs (fostering autotrophic respiration) and the large active layer depth (facilitating heterotrophic respiration).

➢ The estimated $Q_{10}$ values of 1.42 and 1.48 are well within the range of $1.3 \lesssim Q_{10} \lesssim 1.5$, which was retrieved across different ecosystems and climates (Mahecha et al., 2010). Furthermore, the fact that $Q_{10,1}$ was lower than $Q_{10,2}$ is in accordance with a concept, which suggests a correlation between a lower/greater soil temperature sensitivity and a drier/wetter tundra (Olefeldt et al., 2013).

➢ The estimated $P_{max}$ values also follow a seasonal course reflecting the growth and senescence of the canopy. The value of $P_{max,1}$ being greater than $P_{max,2}$ is due to the larger biomass of the bushes relative to the sedges. The values agree well to the maximum assimilation rates of approximately 15.9 $\mu$mol m$^{-2}$ s$^{-1}$ and 11.1 $\mu$mol m$^{-2}$ s$^{-1}$ that are respectively found for *Salix pulchra* and *Carex aquatilis* during the peak of the Arctic growing season (Oberbauer and Oechel, 1989; Tieszen, 1975). Given a mean mid-growing season $P_{max}$ of 8.6 $\mu$mol m$^{-2}$ s$^{-1}$ for northern peatlands, $P_{max,2}$ (8.9 $\mu$mol m$^{-2}$ s$^{-1}$) constitutes a representative uptake capacity, whereas $P_{max,1}$ (12.3 $\mu$mol m$^{-2}$ s$^{-1}$) suggests a comparatively large potential for sequestering carbon dioxide (Frolking et al., 1998; Laurila et al., 2001). Another aspect that indicates the reliability of the estimated $P_{max}$ values is their correlation with the normalised difference vegetation index (NDVI) as seen at many other tundra ecosystems (Mbufong et al., 2014; Shaver et al., 2007). Regarding both growing seasons, the footprint's NDVI was greater in 2015 suggesting a more active vegetation than in 2014 (ORNL, 2017). Similarly, the $P_{max}$ values of both vegetation classes, in particular the values of the more abundant vegetation class 2, were greater during 2015. Satellite records for tundra landscapes are, however, often confounded by various effects that are particularly profound in high-latitude regions (Stow et al., 2004). Therefore, satellite-derived NDVI values of tundra ecosystems may need to be double-checked with optical sampling in the field, if they are applied to resolve interannual differences (Gamon et al., 2013).

➢ The estimated $\alpha$ values amount to 0.042 ($\alpha_1$) and 0.04 ($\alpha_2$), and are thus greater than the mean mid-growing season $\alpha$ of northern peatlands amounting to 0.023 (Frolking et al., 1998; Laurila et al., 2001). The high light sensitivity indicates an efficient physiology enabling a considerable photosynthetic activity at low irradiance levels. A similar ratio between both vegetation classes was found by compiling numerous quantum yields that were obtained during the Arctic peak season: 0.038 for *Salix* spp. and 0.03 for *Carex* spp. (Shaver et al., 2007).



### 4.3. Interpretation of diurnal, seasonal and interannual flux variabilities

On the diurnal scale, the temporal variability in the carbon dioxide fluxes was controlled by meteorological conditions. For comparison among climate-relevant trace gases at this site, the methane fluxes exhibited a larger temporal variability, which was rather governed by the spatial variability, i.e. the constantly varying source area composition in the fetch (Rößger et al., 2018). On the interannual scale, the carbon dioxide fluxes displayed, in contrast to methane fluxes, a larger variability, which was driven by abiotic factors such as snow melt timing and initial growing season temperatures (Aurela et al., 2004; Groendahl et al., 2007). In the case of this heterogeneous area, however, biotic factors such as canopy structure and distribution also provide explanatory power.

In 2015, the rapid snow melt coincided with the spring flood, thus enabling a mutual start of canopy development for both vegetation classes in early June. The growing season was initialised in mid-June by mosses, which are much more abundant in vegetation class 2 (Fig. 8). Mosses are, in contrast to vascular plants, able to start assimilating right after snow melt since their photosynthetically active tissue can be maintained over winter (Oechel, 1976). From this point until late September/early October, mosses formed a basal net uptake. Considerable moss activity until late autumn has also been observed on the nearby river terrace (Eckhardt, 2018; Kutzbach et al., 2007b). Furthermore, mosses can account for distinctly more than half of total photosynthesis as demonstrated for graminoid areas with high moss cover (Douma et al., 2007; Sommerkorn et al., 1999). However, it is possible that mosses did not fully photosynthesise throughout the growing season due to their tendency to lower their photosynthetic capacity under high irradiance (Murray et al., 1993). This light stress depends on cloudiness, sun angle, moss structure and shadowing by vascular plants, altogether promoting a late-season activity of mosses while other plants went already dormant (Zona et al., 2011). On top of the basal moss activity, the shrubs of vegetation class 1 exhibited a larger net uptake until the growing season peak around late July/early August, after which the sedges of vegetation class 2 dominated the carbon dioxide exchange. The fact that *Carex* spp. started growing earlier than *Salix* spp. has also been observed at other sites; however, considerable variation exists in the timing of phenological events both among and within species (Chapin III et al., 1992; Wielgolaski, 2012).

In 2014, air temperatures were higher than the monthly long-term means throughout the measurement period (Fig. 4). During the early and slow snow melt in mid-May, the low sedges and mosses remained buried in the depressions with accumulated snow longer than the large bushes on the elevated ridge with less snow. Thereby, the willow twigs were exposed to daytime temperatures above freezing leading to the development of catkins in late May already. Hence, vegetation class 1 was more advanced in its phenology than vegetation class 2 at the onset of the growing season. The consequence was the substantially larger net uptake of the shrubs until the seasonal peak in early August. Apparently, the shrubs largely benefitted from elevated early growing season temperatures, an effect that has also been found favourable for shrub encroachment in the Arctic (Myers-Smith et al., 2011). Incidentally, shrubs have been growing on Samoylov Island only since the 1960s (Pfeiffer, E.-M., personal communication, 2017). Besides the delayed phenological development, the low carbon sequestration of vegetation class 2 during that period can also be attributed to a soil moisture deficit-induced decline



in net assimilation of mosses as they are prone to desiccation due to both missing roots and the absent ability to actively regulate their internal water content (Turetsky et al., 2012). After the growing season peak and similar to the other year, vegetation class 2 dominated the net uptake, in particular during late August, which can likely be ascribed to enhanced moss
activity.

## 4.4. Appraisal of the budgets' representativeness

The spatial representativeness of the observed fluxes can be assessed with the sensor location bias (Schmid and Lloyd, 1999). If the flux rates of the considered surface classes are similar, as
in 2015, the deviation between the respective surface class compositions in footprint and area of interest plays a minor role (Table 1). In 2014, the sensor location bias came into effect as the flux decomposition unveiled a varying seasonality between both vegetation classes that was concealed in the net signal. In this case, a quantitative comparison of the flux budgets with other sites lacks validity due to a potentially non-representative surface class composi-
tion, i.e. the comparison of the flood plain's greenhouse gas budgets with the budgets of the river terrace must remain restricted to Samoylov Island and cannot be extended on the Lena River Delta (Table 2). The revealing outcome of the flux decomposition proofs its utility for an enhanced interpretation of eddy covariance data by gaining insights into the phenological dynamic of individual vegetation classes. It also demonstrates that climatologically unusual
conditions can adversely affect the representativeness of the footprint, resulting in the potential need to regularly examine the representativeness of apparently homogeneous footprints, in particular during prolonged unusual weather conditions as biased budgets may otherwise be estimated.

The temporal representativeness of the obtained budgets may thus be constrained on the in-
terannual scale. As the air temperatures in 2015 better correspond to long-term means than in 2014, the 2015 budgets are better suited for an inter-site comparison (Table 3). Moreover, the obtained budgets also possess a confined validity on the annual scale since they only cover a period that is similar to the growing season. Outside this period, no uptake of carbon dioxide occurs, implying a lower year-round sink strength for greenhouse gases. This assumption is
based on the accumulating evidence that the release of carbon dioxide and methane is not negligible during the very cold winter – in contrast to the traditional view of a wintertime inactivity in Arctic ecosystems (van der Molen et al., 2007). For instance, at multiple sites in Alaska, the cold season release of carbon was found to equal 1-2 times the warm season net uptake (Euskirchen et al., 2012; Oechel et al., 2014; Zona et al., 2016).
## 4.5. Comparison of the budgets with other Arctic sites

Across various Arctic flux sites, the flood plain of Samoylov Island exhibits a carbon dioxide sink strength being distinctly greater than the average (Fig. 1 and Table 3). This aspect appears noteworthy, when local conditions are taken into consideration: the mean net radiation
during the growing season is lower than for most Arctic sites, and the underlying permafrost displays one of the lowest ground temperatures in the world (Boike et al., 2013; Obu et al., 2018; Romanovsky et al., 2010). The diminishing effects of these climate factors are counterbalanced by the deposition of nutrients in the course of spring flooding (van Huissteden et al.,





2005). Among the three great Siberian rivers draining into the Arctic Ocean (Ob, Yenisei, Lena), the Lena river ranks first in terms of total suspended matter (Cauwet and Sidorov, 1996). A large portion of this matter is transported during the annual spring flood, thereby regularly mitigating the nutrient limitation that affects many Arctic ecosystems (Beermann et al., 2014;
Fedorova et al., 2015).

More specifically, the net uptake of the flood plain on Samoylov Island is distinctly weaker compared to flood plains of the Siberian rivers Kolyma and Indigirka (Kittler et al., 2017; Parmentier et al., 2011). Other Siberian sites encompass Seida and Lavrentiya, which exhibit a similar and stronger net uptake, respectively (Marushchak et al., 2013; Zamolodchikov et al.,
2003). Furthermore, the flood plain's net uptake is considerably stronger than budgets of high Arctic sites in Svalbard, Greenland and Canada (Lafleur et al., 2012; López-Blanco et al., 2017; Lüers et al., 2014; Lund et al., 2012). In comparison to sites in either low Arctic or sub-Arctic, no general conclusions can be drawn, which is likely due to the ubiquitously high spatiotemporal flux variability in the Arctic region. Also, no uniform picture emerges in the com-
parison with Scandinavian peatlands (Aurela et al., 2002, 2009; Fox et al., 2008). When comparing with sites in the northern part of the north slope of Alaska, the flood plain exhibits a substantially stronger net uptake (Oechel et al., 2014; Raz-Yaseef et al., 2017); in the southern part, however, similar net uptakes seem to prevail (Euskirchen et al., 2016).

**5. Conclusion**

The core of the present study are the advanced scaling options of the demonstrated flux decomposition methodology, i.e. fitting a set of area-weighted, surface class-specific flux models to the observed flux. In this way, two major advantages could be gained. Firstly, downscaling net flux signals from the mesoscale to the microscale yielded flux rates for homogeneous land-
scape units, therefore generating valuable insights into seasonal variability and functional flux to flux driver relationships of major tundra vegetation types. Moreover, these unbiased flux rates offer the possibility to aid the calibration of macro-scale models or the validation of their sub-grid variability. Secondly, upscaling the decomposed flux rates to a larger area circumvented the sensor location bias of the study site, and thus yielded defensible flux budgets,
which take the pronounced surface heterogeneity into account. Moreover, the values estimated for the fitting parameters (in particular $P_{max}$) provide the opportunity to contribute to the estimation of carbon dioxide budgets on the macroscale (e.g. pan-Arctic) based on their relationships with remote sensing-derived parameters such as NDVI.

While the aggregated seasonal flux rates of both pre-defined classes (bushes and sedges) were
mostly similar, the flux decomposition revealed a varying seasonality that was hidden in the net signal during a comparatively warm spring period. Accordingly, a seasonal difference between locally observed and regionally estimated fluxes can emerge in response to climatologically unusual conditions. This aspect may gain importance against the projected rise in weather extremes in the course of climate change. Beyond such anomalous situations, the flux de-
composition may also be important in a general context as footprints are frequently assumed homogeneous, but surfaces are seldom entirely homogeneous (depending on the desired scale and the examined greenhouse gas). In this context, the flux decomposition methodology can be adopted in other tundra ecosystems as well as regions outside periglacial environments, and



hence may be supportive in the fields of landscape ecology, experimental agronomy, catchment hydrology and biogeochemical modelling.

*Information on data availability and author contributions will be handed in later.*

## Acknowledgements

The conduct of the present study would not have been possible without the funding from the German Federal Ministry of Education and Research, which enabled the research projects "CarboPerm" (grant number 03G0836A) and "KoPf" (grant number 03F0764A). Further sup-

port was obtained by the Cluster of Excellence "CliSAP" (EXC177), University of Hamburg, funded through the DFG. A warm word of thanks must go to the staff of the research station on Samoylov Island as well as the folks in Potsdam and Tiksi organising transport, accommodation and logistics. Furthermore, a shout-out shall be given to Ben Runkle for the closed chamber data. Cordial thanks must also be expressed to Felix Ament for his advisory support.

The present study is dedicated to travelling all points of sail and savouring the entire colour spectrum.

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





**Figures**

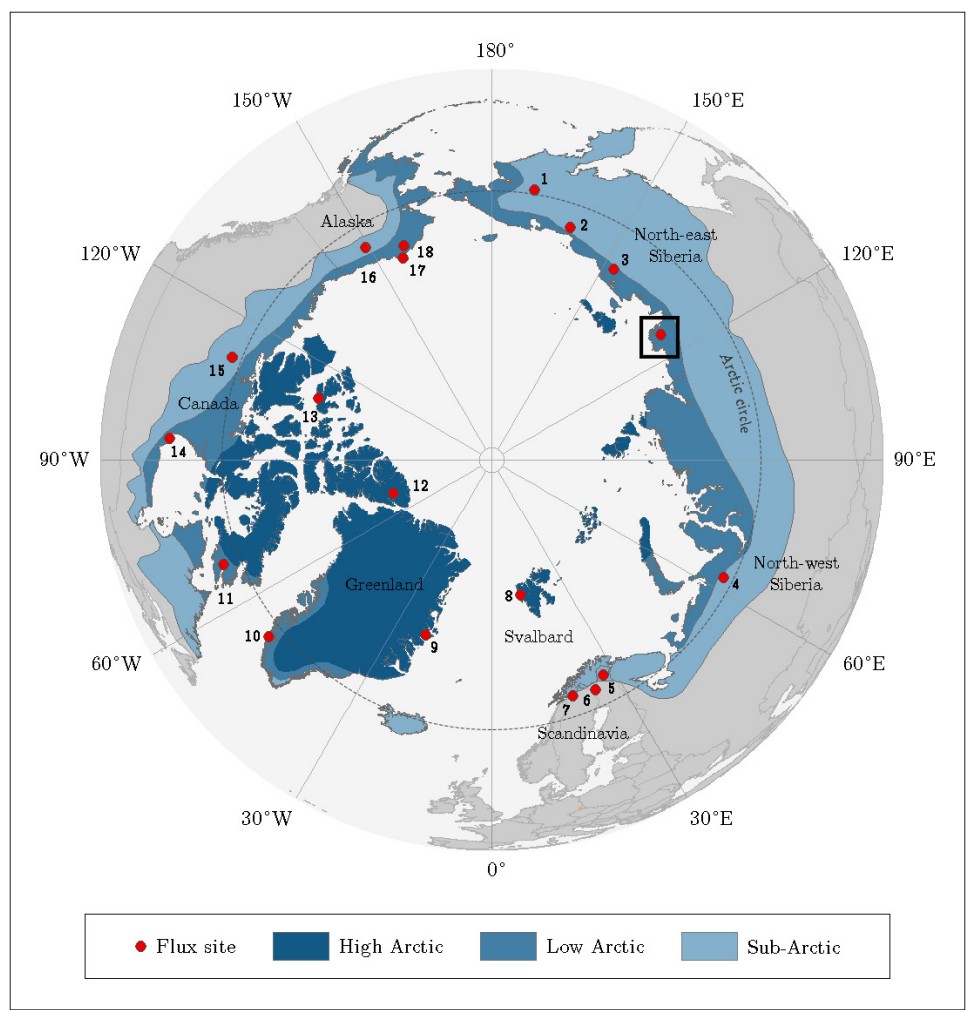

Fig. 1. Location of the Lena River Delta in northern Siberia indicated by the square. The dots
point out sites that were utilised for the pan-Arctic comparison of carbon dioxide budgets
(Table 3). The classification of the Arctic zones was based on vegetation occurrence (modified
from AMAP, 1998). Accordingly, the treeline delimits the (terrestrial) Arctic, i.e. it corre-
sponds with the boundary between sub-Arctic and low Arctic.




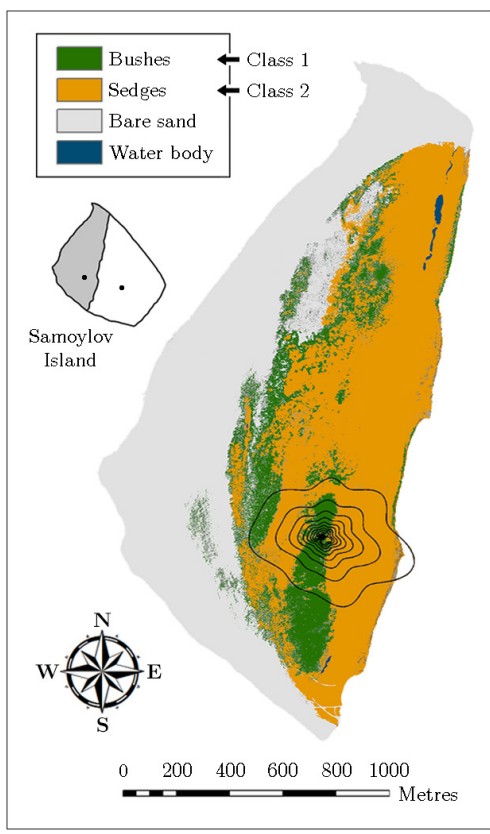

Fig. 2. Vegetation map of the flood plain on Samoylov Island obtained through supervised classification of a high-resolution orthomosaic. The flux tower was situated in the centre of the footprint isolines, which indicate the averaged area from which 10 - 90 % (increment of 10 %) of the flux originated during both measurement periods 2014 and 2015 (footprint climatology). The small inset illustrates Samoylov Island being composed of flood plain (grey) and river terrace (white) plus the location of their respective flux towers.



Fig. 3. Schematic overview of the model calibration, which contains four steps within different parameterisations were applied to obtain significant fitting parameters ($R_{base}$, $Q_{10}$, $P_{max}$, $\alpha$). The values (e.g. 3-2-p model) denote the number of parameters to be fitted.



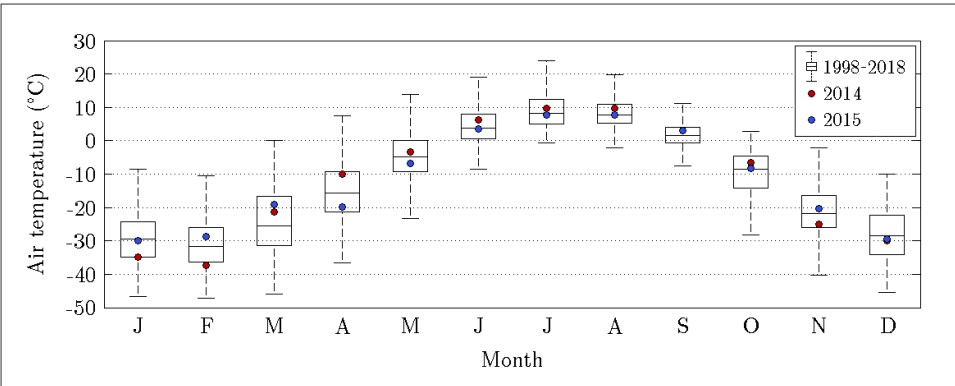

Fig. 4. Annual course of air temperature on Samoylov Island for the years 2014 and 2015 as well as the recent 20-years baseline (Boike et al., 2013, 2018). In each boxplot, the central mark denotes the monthly median, and the bottom and top edges indicate the 25[th] and 75[th] percentiles, respectively. The whiskers extend to the most extreme data points excluding outliers. During the warm season, when flux data was available (June to September), 2014 was mostly warmer than 2015.

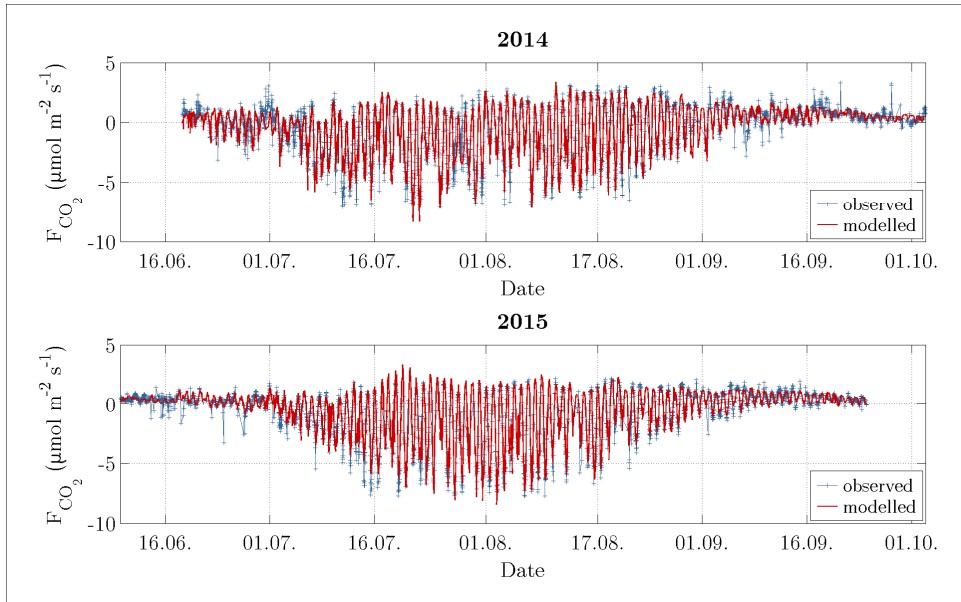

Fig. 5. Time series of observed carbon dioxide fluxes (after conducting the quality assessment) and modelled fluxes. During the growing season, which is indicated by an elevated variability between late June and early September, the daytime uptake directly followed the diurnal cycle of PPFD while the nighttime release was dependent on air temperature.





Fig. 6. Time series of fitting parameters in 2014 for vegetation class 1 (index 1 and green confidence intervals) and vegetation class 2 (index 2 and yellow confidence intervals). The circles represent acceptable fits while the respective reasons for reparameterisation such as insignificance or out-of-valid-range are signified by plus signs and squares. The triangles denote the fitting parameter(s), which caused a refit in the corresponding vegetation class.





Fig. 7. Time series of fitting parameters in 2015 with the same symbols and colours as utilised in the previous figure.





Fig. 8. Time series of decomposed fluxes with 95 % confidence intervals accounting for both vegetation classes. The width of the confidence intervals varied depending on both the flux magnitude and the number of fitting parameters in the chosen model. The decomposition revealed a distinct difference in NEE between both vegetation classes during the first half of the growing season in 2014, while the flux dynamics of both vegetation classes were rather similar during the remaining time.





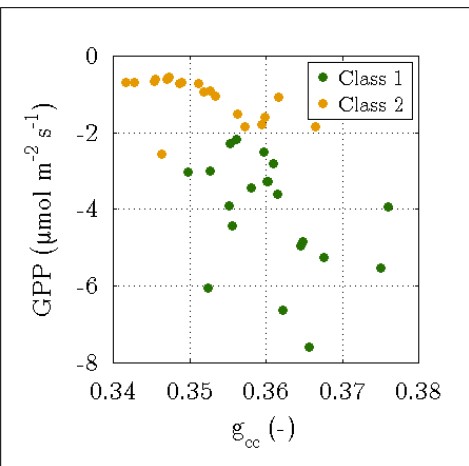

Fig. 9. Daily means of photosynthesis, obtained for both vegetation classes between 18[th] June and 7[th] July 2014, versus their corresponding green chromatic coordinates ($g_{cc}$), acquired from time lapse images of the footprint. Employing these images, the $g_{cc}$ values depict the fraction of the green colour in relation to the three primary colours in the RGB colour space. The significantly greater greenness in vegetation class 1 is associated with larger photosynthetic rates, while vegetation class 2 is characterised by a less green canopy and thus a lower photosynthetic activity ($P < 0.05$).





**Tables**

Table 1. Outcome of downscaling and upscaling carbon dioxide fluxes for the comparison period 18[th] June to 24[th] September in both 2014 and 2015. The mean downscaled fluxes (± standard deviation) refer to the individual fluxes (Fig. 8). Their aggregation yielded cumulative fluxes, whose projection on their corresponding areas on the flood plain in turn returned upscaled fluxes (± combination of cumulative flux error and classification error).

| Vegetation class | Fractional cover on flood plain (m²) | Classification uncertainty (%) | Downscaled $F_{CO_2}$ ($\mu$mol m⁻² s⁻¹) | | | | | | Upscaled $F_{CO_2}$ (Mmol) | | | | | |
| | | | 2014 | | | 2015 | | | 2014 | | | 2015 | | |
| | | | NEE | TER | GPP | NEE | TER | GPP | NEE | TER | GPP | NEE | TER | GPP |
|---|---|---|---|---|---|---|---|---|---|---|---|---|---|---|
| 1 | 251891 | 19.1 | -0.89 ± 2.86 | 2.62 ± 0.84 | -3.51 ± 3.32 | -0.71 ± 2.51 | 1.87 ± 0.99 | -2.58 ± 3.11 | -1.89 ± 0.36 | 5.56 ± 1.06 | -7.45 ± 1.42 | -1.51 ± 0.29 | 3.96 ± 0.75 | -5.47 ± 1.04 |
| 2 | 795065 | 19.5 | -0.38 ± 1.81 | 1.81 ± 0.71 | -2.19 ± 2.21 | -0.69 ± 2.32 | 2.11 ± 0.68 | -2.81 ± 2.62 | -2.53 ± 0.33 | 12.12 ± 1.55 | -14.65 ± 1.87 | -4.66 ± 0.59 | 14.11 ± 1.81 | -18.77 ± 2.41 |




Table 2. Comparison of the sink/source strengths between flood plain and river terrace for the comparison periods in 2014 and 2015 (Holl et al., 2018; Rößger et al., 2018). Accounting for the methane's radiative efficiency as a potent greenhouse gas, the methane budgets were converted to carbon dioxide equivalents with a factor of 34, which corresponds to methane's global warming potential based on a time horizon of 100 years including climate carbon feedbacks (Myhre et al., 2013). The flood plain budgets are given for each vegetation class and for the total area. These budgets are the result of a scaling procedure, which included fairly large classification errors that caused distinctly greater uncertainties in comparison to the river terrace budgets, which derived from a representative footprint and hence did not undergo any scaling processes. In comparison to the flood plain, the polygonal tundra on the river terrace took up less carbon dioxide per square metre, but also released less methane resulting in a similar (2014) and weaker (2015) sink strength for greenhouse gases.

| Geo-morphological unit | Vegetation class | $F_{CO2}$ (mol $CO_2$ m$^{-2}$) | | $F_{CH4}$ (mol $CH_4$ m$^{-2}$) | | Greenhouse gases (mol $CO_2$ eq m$^{-2}$) | |
|---|---|---|---|---|---|---|---|
| | | 2014 | 2015 | 2014 | 2015 | 2014 | 2015 |
| Flood plain | 1 | -7.51 ± 1.43 | -5.99 ± 1.15 | 0.004 ± 0.001 | 0.002 ± 0.001 | -7.45 ± 1.43 | -5.98 ± 1.15 |
| | 2 | -3.18 ± 0.42 | -5.86 ± 0.74 | 0.213 ± 0.042 | 0.221 ± 0.042 | -0.55 ± 0.66 | -3.12 ± 0.91 |
| | Total | -4.22 ± 0.47 | -5.89 ± 0.63 | 0.163 ± 0.032 | 0.169 ± 0.032 | -2.21 ± 0.61 | -3.81 ± 0.74 |
| River terrace | Total | -3.47 ± 0.03 | -3.74 ± 0.03 | 0.096 ± 0.001 | 0.099 ± 0.001 | -2.29 ± 0.03 | -2.52 ± 0.03 |



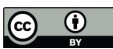

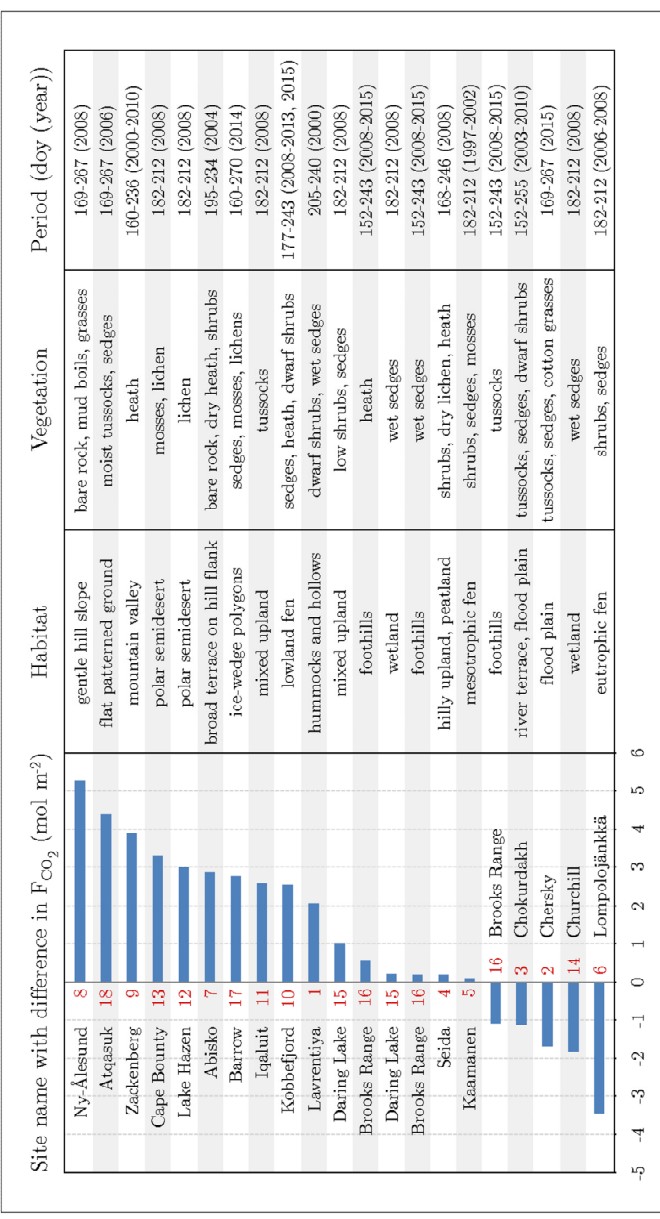

Table 3. Putting the carbon dioxide budgets of the flood plain in perspective with budgets estimated by means of the eddy covariance method at other Arctic sites. A negative/positive difference in the budgets indicates a carbon dioxide sink strength of the comparison site being greater/lower than the one of the flood plain on Samoylov Island. For an appropriate comparison, the budgets of this study's site were recomputed for the days of the year (doy) periods of the other sites. In this process, representative 2015 flux data was basically utilised, unless the budget of the other site derived from 2014. If the budget period of the other site exceeded the period of this study's budget, the (few) missing days in the start/end were filled with the daily sum of the first/last day in the budget period. The numbers next to the bars refer to the numbers in the circumpolar map (Fig. 1). Compared to various Arctic sites, the flood plain on Samoylov Island constitutes a distinctly strong carbon dioxide sink.





## Appendix

The modelling of carbon dioxide fluxes was based on a fitting procedure that comprised four steps (Fig. 3).

Step №1: The procedure first fitted the final model to the observed fluxes of each day utilising
a moving window with a fixed size of 14 days. The choice of a suitable window size was based on identifying an optimum between two conflicting demands: the window size ought to have been as small as possible to capture most of the variability in fluxes, whereas the window size ought to have been as large as possible to obtain less noisy time series of preferably significant values of the eight fitting parameters. In general, the target of each fitting process comprised
the estimation of reasonable seasonal courses of meaningful and non-negative best-fit values, whose 95 % confidence interval did not overlap zero. Running the model with varying window sizes and counting the number of significant values for each model run revealed the following: increasing the window size caused the number of significant values to rise, soon to level off, and eventually at a window size of 14 days, to remain at similar values. Prior to calibrating
the model, the following requirements needed to be satisfied for every window: flux samples were available for at least 80 % of the period, the air temperature spread did not fall below 12 °C, and the mean air temperature did not drop below -10 °C. Imposing these requirements was meant to ensure robust and representative fits.

The purpose of this step involved the fixation of $Q_{10}$ in order to prevent overparameterisation,
and moreover, alterations in temperature sensitivity were thought to be less plausible and hence expected to be negligible. This assumption was confirmed by the time series of estimated $Q_{10}$ values displaying an implausible variability, whereas the other fitting parameters presented a rather seasonal course. Based on the deliberation of negligible alterations in temperature sensitivity during both years, the model was run for 2014 and 2015 together during this
step, whereas the model was respectively run for the measurement periods 2014 and 2015 during the next steps. The application of a larger period provided more data points, with the aid of which $Q_{10}$ could be fixed at a more representative value. Two final $Q_{10}$ values were determined for each vegetation class by calculating the median out of all estimated best-fit $Q_{10}$ values, which met the following two requirements: statistical significance and an associated coeffi-
cient of determination of at least 0.75 between observed NEE and modelled NEE.

Step №2: The model was run with $Q_{10}$ being fixed throughout the measurement periods 2014 and 2015 applying a moving window with a fixed size of 14 days and a step size of 1 day again. The requirements laid down in the previous step prior to fitting had to be met again except the requirement of a sufficient air temperature spread.

The aim of this step comprised the creation of two replacement functions for α after six best-fit parameters were estimated. The necessity for replacement functions arose through large peaks in the time series of α. These peaks tended to occur at the onset of the growing season and were hence deemed spurious. Large α values would have promoted photosynthesis, which was of rather minor magnitude at that time of the year. In order to reproduce the low ob-
served NEE, the erroneously elevated GPP was counteracted by a mistakenly enhanced TER



utilising a large $R_{base}$. The resulting problem of equifinality would thus hamper the interpretation of the fitting parameters.

To prevent this adverse circumstance, the two replacement functions, one for each vegetation class, were calculated by fitting a Gaussian bell curve to the time series of significant $\alpha$ values.
In addition, two threshold functions were computed for each replacement function by adding/subtracting 30 % of the function values to/from the replacement function. Hence, the threshold functions formed an interval around the replacement function within the estimated $\alpha$ values were accepted during the further procedure. The threshold of 30 % was visually selected since this value generated an interval, outside of which only the peaks were situated, i.e. spu-
rious and meaningful $\alpha$ values could be reliably separated.

Step №3: The model was initially run with the same parameterisation as in the previous run, but employing a moving window with a flexible size for every day. The application of a flexible window allowed a closer reproduction of the variability in the observed data through adjusting its size. However, since small windows were in conjunction with a small amount of flux
samples, which increased the risk of estimating insignificant parameters, every fit required a minimum of 240 flux samples, which equals 5 days with 48 fluxes per day. Based on this setting, the model was run and the estimated parameters were checked for significance. If one best-fit parameter was insignificant, the window size was increased by one day and the model was run again. This procedure was repeated until a maximum window size of 20 days, if all
fitting parameters were not significantly estimated before utilising a preferably smaller window size.

The objective of this step included the bulk of model calibration within the fitting procedure. Hence, after the initial model run of this step, its output was inspected in two respects: the significance of the remaining fitting parameters ($R_{base}$, $P_{max}$, $\alpha$) and the location of $\alpha$ (inside or
outside the acceptance interval). In case of all six fitting parameters being significant and the two fitted $\alpha$ were situated between the respective thresholds, the estimated NEE was accepted and appended to the modelled time series. If one criterion/both criteria was/were not fulfilled, another model with less parameters was run employing, again, a moving window with a flexible size for every day. This simplification comprised the application of $\alpha$ values adopted from
the previously defined replacement function. The model choice depended on the vegetation class, where the criterion/criteria was/were not satisfied. Hence, $\alpha$ values from the replacement function were employed for either one or both vegetation classes. For instance, if the fitting parameters of only one vegetation class were insignificant, only this vegetation class was refitted applying a replacement $\alpha$ whilst reutilising the retained significant fitting parameters of
the other vegetation class. Subsequently, the significance of the re-fitted parameters was examined. If all parameters were significant, the correspondingly estimated NEE was added to the modelled time series. Any remaining insignificances were otherwise addressed in the next step.

In preparation for the next step, two replacement functions for $P_{max}$ were created. This fitting parameter was chosen over $R_{base}$ since $P_{max}$ featured more insignificant values than $R_{base}$. Once
again, a Gaussian bell curve was fitted to the time series of significant $P_{max}$ values of each vegetation class.



Step №4: Towards the end of the procedure, a greatly simplified model was run for each day applying a moving window with a fixed size. This size corresponded to the average of all window sizes found during the previous step.

The goal of this step encompassed the remaining model calibrations for a complete time series
5  of modelled NEE. To achieve this target, the model included only three best-fit parameters: $R_{base}$ twice and $P_{max}$ once. The second $P_{max}$ for the other vegetation class was adopted from its previously calculated replacement function. This confined parameterisation was, given a constant amount of observed flux samples, associated with an elevated number of degrees of freedom, which in turn allowed a more precise estimation of the remaining fitting parameters, i.e.
10  their confidence intervals were smaller. In this way, all best-fit parameters were significant and could be utilised for a reliable modelling of NEE.