# Peer review of "Scaling and balancing carbon dioxide fluxes in a heterogeneous tundra ecosystem of the Lena River Delta"

_Biogeosciences, 2019_

## Referee Comment (RC1) · Anonymous Referee #1 · 12 Mar 2019

General comments The authors of "Scaling and balancing carbon dioxide fluxes in a heterogeneous tundra ecosystem of the Lena River Delta" have collected what appears to be a robust eddy flux data set from a remote location in the Siberian arctic. They conduct a modeling analysis in order to downscale the fluxes and distinguish between the signals associated with the two types of vegetation cover present in the towers foot-print. They then use the resulting model to calculate a robust greenhouse gas budget. Overall the data, analysis and questions seem to be relevant and of high quality. However, I have a number of questions and suggestions. A more thorough exploration of literature focused on similar questions and utilizing similar methods could improve the introduction and framing of the manuscript. I also have questions regarding why the

authors chose to fit parameters at a daily time step rather than explicitly represent leaf area index or NDVI in the model and why they chose to make comparisons to the parameters and fit statistics of Shaver et al 2007 who utilized a distinctly different model. Moreover, chamber flux measurements, ground truthing of the vegetation classification used in processing the fluxes, methane flux measurements and a correlation between the model parameters and NDVI are mentioned in the discussion but do not appear to be included in the text of the methods or results. Finally the phrasing and word choice at times makes the manuscript challenging to read and interpret. The results and overall analysis seem to be of high quality but the manuscript itself needs changes to clarify some ambiguities and better guide the reader.

Specific comments P2 L3: the meaning of "exposed land area" is unclear. P2 L35: The meaning of "flux variability" is unclear. I'm assuming it refers to variability across space rather than say across time. Also, I would not consider "vegetation composition and structure" to be "environmental controls". P2 L38 – P3 L5: Could this list of the drawbacks of chamber measurements be simplified? A lot of space is dedicated to it and it seems ancillary. Although the authors decompose eddy flux data to make inference about different microforms within a single site I'd argue they do not directly compare eddy flux and chamber measurements and attribute differences between the carbon budgets obtained to any of these particular drawbacks. P3 L10: Clarifying the meaning of "heterogeneous" when it is first used or replacing it with a more precise term could be beneficial. Does it refer exclusively to heterogeneity in vegetation cover? P3 L24: Clarifying the term "microforms" when it is first used and explaining it in relation to the "heterogeneous surface" could also be beneficial. P3 L28-35: The first and second objectives seem to overlap a bit. P3 L28-35: Given that the goals of this study relate to better understanding the impact of heterogeneity in the landscape on fluxes text should be added to the introduction which better makes the case that the patchy nature of tundra ecosystems contributes to the uncertainties in tundra flux budgets and explores issues related to scaling flux measurements across the landscape. A number of articles have addressed questions similar to those asked here, albeit using different methodology, but the text is not framed in relation to many of them (e.g. Shaver et al 2007, Loranty et al 2011 Ecosystems, Kade et al 2012 in journal of geophysical letters, Shaver et al 2013 Philosophical transaction of the Royal Society, Sweet et al 2015 in Global Change Biology). In particular, Shaver et al 2007 is mentioned in the introduction, but its main conclusion that leaf area index explains a large proportion of the variability in $CO_2$ flux across vegetation types isn't explored in detail. P3 L28-35: The introduction could also benefit from discussing the impacts of climate on tundra vegetation and the tundra landscape. Permafrost carbon is an important player in determining the future carbon budget of tundra, but not the only player. This seems especially important given that this article focuses on the impact of vegetation and the physical characteristics of the landscape on fluxes. P5 L9-36: Why is an evaluation of the classifications accuracy using either independent ground-based measurements or digitization not presented in the text (some uncertainty metrics appear in table 1 but aren't mentioned in the methods)? P6 L23: Why was 1m resolution used here when the classification available was of a much higher resolution? P6 L36 - P7 L24: Could this method have been simplified by including information about NDVI/leaf area index or using a smooth function to describe the change in these parameters over time thereby removing the need to estimate so many parameters at a daily time step? Using information about leaf area index might also make the parameterization more generalizable (see Shaver et al 2007,2013). Along these same lines given that the model doesn't include NDVI or leaf area index was the distribution of NDVI/leaf area index for the vegetation in the tower footprint compared to that of the rest of the study area to ensure the model is representative? P10 L30-L35: The river terrace carbon flux budgets and decomposition of the methane fluxes don't appear to be described in any detail in the methods section. P11 L23-24: Although they're mentioned here no ground-truthing methods or results are presented in the text. P11 L37- P12 L2: Why is this described here rather than in the methods and results section? P12 L12: Shaver et al 2007 use a different model, which includes leaf area index and compares chamber flux data from a number of plots in Alaska and Sweden. I don't think it is valid

to compare the fit statistics like this. P12 L21-35: These chamber flux measurements and the sampling design used isn't described in the methods or results section. Do five measurements refer to five individual discontinuous measurements? Moreover, the critique of chamber-derived fluxes seems ancillary to the question of validating the model especially given that the authors seem to be comparing an extremely small number of respiration only chamber measurements to their results. P13 L22-25: A correlation between Pmax and NDVI is mentioned here but doesn't appear to be included in any of the figures or results. P13 L33-L39: Is this comparison valid? Again the model used in shaver et al 2007 represents the canopy in a different way than equation 2 which is effectively a big leaf model (see Rastetter 1992). Also, the units of Pmax and other parameters aren't provided in the methods for equation 2 and 3. P16 L30-33: Again a relationship between the flux parameters and NDVI doesn't appear to be presented in the text or figures. Figure 2: Putting imagery in the background of this figure and including an inset map showing the site and its extent in regional context could be beneficial for the reader. Figure 6,7: Why are not units given for the $\alpha$1,2 axes. Figure 9: It's not clear why a scatter plot is used here if the goal is to show the differences between the two vegetation types. Maybe two bar or box plots would fit better.

Technical corrections P2 L13-16: Suggest simplifying/rephrasing this sentence, in particular, the phrasing of "However, due to ambiguous results and large confidence intervals, it currently remains unclear". P1 L16: Suggest rewording "flux signal associated therewith could extensively be decomposed" P1 L18-19: Suggest rewording "unveiled a differing seasonality" P1 L25-26: Suggest rewording "approved the reliability" P1 L 36-38: Suggest restructuring the final sentence of the abstract. P2 L6: Suggest rewording "historical carbon sink function". P2 L11: Suggest rephrasing "The arctic north of 60° N" P2 L25: Should "ties" read "uncertainties"? P2 L27: Suggest rephrasing "reduction of these discrepant uncertainties". P2 L33, P3 L17 and elsewhere: "aggravates" doesn't seem to fit in this context P3 L21: typo "and the study" -> "and to study" P4 L31: typo "an quarter" -> "a quarter" P5 L11-12: Suggest rephrasing "a very high spatial information density" P5 L13: Suggest listing the software used to

carry out the classification here. P5 L16: Suggest using "shrubs" in place of bushes, as this is more consistent with language used in other literature (see: CAVM Team. 2003. Circumpolar Arctic Vegetation Map). P6 L8-9: Suggest simplifying or rephrasing "respiration multiplies/divides when the temperature rises/drops" P8 L3-5: Suggest simplifying the sentence listing the air temperatures and precipitation rates. P8 L18: The phrasing of "a dominating respiration" is awkward. P8 L24-26: Suggest rephrasing "ripening phase", "verged on full maturity" and "colouration and shedding of leaves". P8 L34-L35: Referring to the classifications by the more descriptive names given earlier rather than a numbing scheme would make the results easier to follow. P9 L16: Suggest rephrasing "featured most of the significant differences between each other" P9 L19: Suggest rewriting/simplifying "On account of both the coinciding varibailites of explanatory variables and explained variable" P9 L25: Suggest rephrasing "less good" P10 L16: Suggest rephrasing "despite methane's minor percentage of roughly 3% in the entire greenhouse gas exchange". P12 L24: Suggest rewriting "mean carbon dioxide flux with a standard deviation of 2.1 +- 0.9" to make it clear that the first number therein is the mean. P14 L11: Suggest rephrasing "mutual start" P15 L17: "proofs" -> "proves" Figures 1-9: The axis and legends provided seem small even when printed on a full page.
* * *

---

## Referee Comment (RC2) · Anonymous Referee #2 · 16 Mar 2019

The paper "Scaling and balancing carbon dioxide fluxes in a heterogeneous tundra ecosystem of the Lena River Delta" introduces new experimental results in estimation of carbon fluxes of tundra ecosystems in Lena river delta (Russia). It is known that the large areas of Northern Eurasia near the Arctic cycle are still very poorly investigated in respect of both spatial and temporal variability of GHG exchange and contribution of different plant communities into global atmospheric GHG budget. It makes the results of the study very interesting for scientists working in ecology, biogeochemistry and micrometeorology. The paper is well written. It contains detailed descriptions of experimental site, design of field experiments, developed model algorithms. Discussion chapter includes close examination of obtained results. Before publishing however

several points of the paper should be clarified.

1. The chapter 2.3 "flux processing" has not information about procedure or method that has been used for gap filling. The percentage of gaps in flux time series is not quantified.

2. The LAI ranges of different vegetation classes should be indicated in chapter describing the surface and vegetation structure. Information about surface topography should be also presented.

3. Figure 2 illustrates the vegetation map of the flood plain on Samoylov Island and shows the tower location. The tower is situated close to the boundary between bushes and sedges. They have different height and, probably, different density. It can be expected that the air flow disturbances at the boundary between these vegetation types can influence the wind and turbulence patterns at tower location and as a result the measured fluxes taking into account the height of eddy covariance equipment installation (2.8 m).

4. The photos of Samoylov Island, that can be found in Internet, show a very nice landscape and, at the same time, a non-uniform surface topography of the study area. Did you estimate the possible effects of non-uniform surface topography on measured fluxes? I guess the possible uncertainties in flux estimation due to complex topography should be discussed in the paper.

Specific comments. Page 8 line 3 " The mean air temperatures during the measurement periods in 2014 and 2015 ..." I guess the periods of flux measurements have to be indicated in the paragraph... e.g. from June to October 2014 and from June to September 2015.

Page 11 Line 8-10 "While the entire temperature sensitivity of NEE is manifested through changes in TER, the effect of temperature on the biochemical reactions in GPP is neglected (Haraguchi and Yamada, 2011)." I' m not sure that it is a very good

assumption for accurate NEE parameterization. It is well known that GPP is strictly depended on temperature and the influence of air temperature changes on GPP rate is actually comparable with effect of temperature changes on TER.

Page 11 line 19 ... direct and diffuse solar radiation ...

Page 14 line 1 ... seasonal and interannual carbon flux variability ....

Page 14 line 20-22 "However, it is possible that mosses did not fully photosynthesize throughout the growing season due to their tendency to lower their photosynthetic capacity under high irradiance ". What is the reason of such effect? May be it is the result of moss overheating and deficiency of internal water content?

Page 14 line 22-23 What is it, "sun angle"? Do you mean sun elevation ?
* * *

---

## Author Comment (AC1) · 29 Apr 2019

We sincerely thank the reviewer for providing his helpful comments, which we implement in order to improve the quality of our manuscript. Please find our response in the attached pdf file.

Please also note the supplement to this comment:
https://www.biogeosciences-discuss.net/bg-2019-10/bg-2019-10-AC1-supplement.pdf

---

## Author Comment (AC2) · 29 Apr 2019

**Reply to comments of referee 2**

Manuscript:    Scaling and balancing carbon dioxide fluxes in a heterogeneous tundra ecosystem of the Lena River Delta

Authors:    Norman Rößger, Christian Wille, David Holl, Mathias Göckede, Lars Kutzbach

Colour legend:  Referee's comments are written in red
Our responses are written in blue

**General comments**

The paper "Scaling and balancing carbon dioxide fluxes in a heterogeneous tundra ecosystem of the Lena River Delta" introduces new experimental results in estimation of carbon fluxes of tundra ecosystems in Lena river delta (Russia). It is known that the large areas of Northern Eurasia near the Arctic cycle are still very poorly investigated in respect of both spatial and temporal variability of GHG exchange and contribution of different plant communities into global atmospheric GHG budget. It makes the results of the study very interesting for scientists working in ecology, biogeochemistry and micrometeorology. The paper is well written. It contains detailed descriptions of experimental site, design of field experiments, developed model algorithms. Discussion chapter includes close examination of obtained results. Before publishing however several points of the paper should be clarified.

We thank the reviewer for the comments, which helped us to improve our manuscript.

1. The chapter 2.3 "flux processing" has not information about procedure or method that has been used for gap filling. The percentage of gaps in flux time series is not quantified.

There was no classic gap-filling that involved filling gaps in a time series of observed values with modelled values. The reason is that we were interested in the flux budgets of our area of interest (i.e. the entire flood plain) rather that the footprint (i.e. only a part of the flood plain). For this purpose, we only used modelled (instead of a mix of observed and modelled) values to estimate the final budgets given in Table 3. Therefore, the percentage of gaps in the time series (roughly 40 %) was of less importance.

2. The LAI ranges of different vegetation classes should be indicated in chapter describing the surface and vegetation structure. Information about surface topography should be also presented.

We did not characterise the vegetation classes with the LAI parameter as LAI values are not available for this site. In fact, we conducted multiple measurements with a LAI-2200C Plant Canopy Analyzer in 2015, but unfortunately the results were not satisfying. The problem was that this kind of measurement is not applicable for such little biomass (sedges, mosses, lichen) as at our tundra site. It was not possible to create a discrete time series of objective measurements over the growing season, as the LAI measurement strongly depended on the position of the device. Consequently, the LAI values largely varied while sampling the vegetation at the same spot.

The topography around the flux tower exhibited a slightly undulating relief ranging from 7.8 m to 10.7 m above sea level. This ancillary information is available in Rößger et al. (2019), which is referenced for further information in section 2.4. The essential information, however, are given directly in the text. Please see "…a sandy ridge aligned in the north-south-axis. The elevated area…" and "…located in depressions around the dry ridge…".

3. Figure 2 illustrates the vegetation map of the flood plain on Samoylov Island and shows the tower location. The tower is situated close to the boundary between bushes and sedges. They have different height and, probably, different density. It can be expected that the air flow disturbances at the boundary between these vegetation types can influence the wind and turbulence patterns at tower location and as a result the measured fluxes taking into account the height of eddy covariance equipment installation (2.8 m).

The vegetation cover of the flood plain on Samoylov is heterogeneous, which served as a starting point for the flux decomposition in this study. As far as the air flow is concerned, our tundra site constitutes – in comparison to other ecosystems or urban sites – a fairly homogeneous site, thereby fulfilling the eddy covariance assumption of a smooth and uniform surface. This statement is based on scrutinising the turbulence with the aid of the integral turbulence characteristics (ITC): during 93 % of the measurements, a well-developed turbulence was present. The remainder accounts for periods, when the wind approached the anemometer (CSAT3) from the back, causing a self-sheltering effect. Consequently, the possible air flow disturbances at the boundary between bushes and sedges could not be registered at the flux tower.

4. The photos of Samoylov Island, that can be found in Internet, show a very nice landscape and, at the same time, a non-uniform surface topography of the study area. Did you estimate the possible effects of non-uniform surface topography on measured fluxes? I guess the possible uncertainties in flux estimation due to complex topography should be discussed in the paper.

We disagree that our site is characterised by a "complex topography" as the flood plain has a slightly undulating relief, shaped by the annual spring flood that has been smoothing the topography for a long time. On the other hand, the flood plain is not fully level and has a vegetation cover with a varying roughness. These aspects impact on the data quality. However, we performed a data quality assessment, which included an ITC test plus a stationarity test, with the result that the assumptions of the eddy covariance theory were fulfilled for the very most part (and the fluxes were thus hardly biased by the topography and the varying surface roughness). What we expect from both the slightly undulating relief and the varying vegetation roughness is an increased uncertainty in the footprint modelling. However, the employed footprint model includes a wind direction-dependent set of roughness lengths, and moreover, it is a widely applied tool within the flux community (see p.11 l.24)

**Specific comments**

Page 8 line 3 " The mean air temperatures during the measurement periods in 2014 and 2015 ..." I guess the periods of flux measurements have to be indicated in the paragraph... e.g. from June to October 2014 and from June to September 2015.

The sentence states that these mean air temperatures respectively refer to the measurement periods, whose lengths are defined in section 2.2. We think that these information are not important enough to be repeated as, moreover, the measurement periods cover a very similar period of the year anyway.

Page 11 Line 8-10 "While the entire temperature sensitivity of NEE is manifested through changes in TER, the effect of temperature on the biochemical reactions in GPP is neglected (Haraguchi and Yamada, 2011)." I' m not sure that it is a very good assumption for accurate NEE parameterization. It is well known that GPP is strictly depended on temperature and the influence of air temperature changes on GPP rate is actually comparable with effect of temperature changes on TER.

Indeed, in the nature, the air temperature impacts on NEE through affecting both TER and GPP. For estimating NEE with our model, however, the impact of air temperature on GPP is neglected. We have modified the text in order to make clear,

that the missing effect of air temperature on GPP concerns our model – not the reality in nature.

Page 11 line 19 ... direct and diffuse solar radiation ...

This change was performed as suggested.

Page 14 line 1 ... seasonal and interannual carbon flux variability ....

For specifying the type of flux in that section headline, we had to write "carbon dioxide flux" instead of the suggested "carbon flux" (in order to exclude methane fluxes). However, the focus of this study clearly is the carbon dioxide flux; so we deem that the reader automatically refers to carbon dioxide flux (variabilities) when reading this headline similar to other headlines including the word "flux" such as "2.3. Flux modelling", "3.2. Dynamics of observed fluxes" or "4.2. Validation of the decomposed fluxes".

Page 14 line 20-22 "However, it is possible that mosses did not fully photosynthesize throughout the growing season due to their tendency to lower their photosynthetic capacity under high irradiance ". What is the reason of such effect? May be it is the result of moss overheating and deficiency of internal water content?

A good explanation for the reason of photoinhibition can be found in Zona et al., 2011:

"Drying and high temperature could decrease moss photosynthesis (Oechel and Sveinbjörnsonn 1978, Murray et al. 1989a). However several experiments showed that mosses are generally not water stressed in wet tundra ecosystems in the high Arctic, as their bases are embedded in a peat that tends to retain water (Oechel and Collins 1976, Hickleton and Oechel 1977, Harley et al. 1989, Murray et al. 1989a). Moreover, elevated temperature generally is less damaging than high irradiance as the temperature optima for photosynthesis of mosses often exceed ambient temperatures in the Arctic and the temperature optima adjust through rapid acclimatization (Oechel 1976, Oechel and Collins 1976, Harley et al. 1989). While bryophytes can adjust to temperature, they cannot acclimate to high light due to the low nitrogen (N) levels in their tissues (Clymo and Hayward 1982, Murray et al. 1993). N deficiency inhibits the protein synthesis necessary to recover from photochemical damage (Ohad et al. 1984, Murray et al. 1993, Huang et al. 2004). In fact, a decrease in photosynthesis due to high irradiance has been observed in mosses even under optimum temperature and fully hydrated conditions (Oechel and Collins 1976, Harley et al. 1989, Murray et al. 1993, Hajek et al. 2009)."

Page 14 line 22-23 What is it, "sun angle"? Do you mean sun elevation?

Yes! The word has been modified.

**References**

Rößger, N., Wille, C., Veh, G., Boike, J. and Kutzbach, L.: Scaling and balancing methane fluxes in a heterogeneous tundra ecosystem of the Lena River Delta, Agric. For. Meteorol., 266-267, 243-255, https://doi.org/10.1016/j.agrformet.2018.06.026, 2019.

Zona, D., Oechel, W. C., Richards, J. H., Hastings, S., Kopetz, I., Ikawa, H. and Oberbauer, S.: Light-stress avoidance mechanisms in a Sphagnum-dominated wet coastal Arctic tundra ecosystem in Alaska, Ecology, 92(3), 633–644, https://doi.org/10.1890/10-0822.1, 2011.

---

## Author Response (AR1)

**Universität Hamburg**
DER FORSCHUNG | DER LEHRE | DER BILDUNG

**Dr. Norman Rößger**
University of Hamburg
Institute of Soil Science
Allende-Platz 2
20146 Hamburg
Germany

Mail: norman.roessger@uni-hamburg.de
Tel: +49 40 42838 4397
Fax: +49 40 42838 2024

**Author's response**

Dear Ito sensei,

I am glad, you concluded - after reading our comments - that we were going to revise the manuscript in a satisfactory manner. We are now done with this and hope that we certainly did so.

Throughout the text, we made many modifications. However, none of these fundamentally changed the content of the text. The most important changes refer to the requests of the referees. They include an extension of the introduction (classification parameters, classification scales, impact of climate change on tundra ecosystems) and the experimental setup (impractical LAI measurements). The remaining alterations in the text are minor and meant to improve both readability and intelligibility. Most of these minor changes were requested by the referees, the other ones were the result of our intention to thoroughly enhance wording/phrasing. In addition, some references, which were in discussion during the initial manuscript submission, were refreshed. You can find all of the conducted changes in the marked-up manuscript.

Please let me know if you have any questions.

I look forward to hearing from you.

Best regards,

Norman Rößger

[revised manuscript text omitted]